# Changes in geometrical aspects of a simple model of cilia synchronization control the dynamical state, a possible mechanism for switching of swimming gaits in microswimmers

**Evelyn Hamilton, Pietro Cicuta** *

Cavendish Laboratory, University of Cambridge, Cambridge, United Kingdom

* pc245@cam.ac.uk

## Abstract

Active oscillators, with purely hydrodynamic coupling, are useful simple models to understand various aspects of motile cilia synchronization. Motile cilia are used by microorganisms to swim and to control the flow fields in their surroundings; the patterns observed in cilia carpets can be remarkably complex, and can be changed over time by the organism. It is often not known to what extent the coupling between cilia is due to just hydrodynamic forces, and neither is it known if it is biological or physical triggers that can change the dynamical collective state. Here we treat this question from a very simplified point of view. We describe three possible mechanisms that enable a switch in the dynamical state, in a simple scenario of a chain of oscillators. We find that shape-change provides the most consistent strategy to control collective dynamics, but also imposing small changes in frequency produces some unique stable states. Demonstrating these effects in the abstract minimal model proves that these could be possible explanations for gait switching seen in ciliated micro organisms like *Paramecium* and others. Microorganisms with many cilia could in principle be taking advantage of hydrodynamic coupling, to switch their swimming gait through either a shape change that manifests in decreased coupling between groups of cilia, or alterations to the beat style of a small subset of the cilia.

## Introduction

Active oscillators coupled through a fluid are an interesting conceptual physical model for an extensive range of systems where liquid is present, mediating interactions between motile cilia [1, 2]. An important application for physics of oscillators in fluids is in understanding the origin of swimming, a property exhibited by water borne organisms across the spectrum of sizes, from large mammals down to small invertebrates and microorganisms like bacteria and unicellular algae [3]. The collective behaviour of larger swimmers, like fish, is thought to be the result of some judgement process of the animal, and controlled neurologically. In contrast,

**Data Availability Statement:** All relevant data are within the paper.

**Funding:** PC was supported by EU ERC CoG HydroSync and EH by Cambridge Trusts. The funders had no role in study design, data collection and analysis, decision to publish, or preparation of the manuscript.

micro-organisms are forced to exploit more basic physical interactions, such as hydrodynamic forces coupling them via the fluid, in order to exhibit emergent behaviour like entraining nutrients in the surrounding [4], optimising flows for nutrition [5, 6], accumulating towards surfaces [7], or self-organising [8–12]. When swimming at the micron length scale, it is the viscous forces which dominate over inertial contributions. This produces some counter-intuitive restraints on their motion [8, 13]: since in the low Reynolds' number (Re) regime the speed at which different configurations are passed through is irrelevant, complex conformation changes are required to impart a net momentum transfer to the fluid over a recurring set of movements. This result is often referred to as the 'Scallop Theorem', in reference to Purcell's famous example of an animal that would be unable to swim at low Reynolds' number [13].

A range of strategies have evolved to tackle the constraint of swimming at low Re. One theme common to many micro-organisms is the development of thin actively moving hair-like filaments protuberances, called motile cilia (or sometimes flagella). These filaments change shape periodically, by bending all along the filament (cilia waveform), and are used to swim. The highly conserved nature of the ultrastructure of cilia within all eukaryotes that express these organelles means many studies on the basics of cilia waveforms and their interactions have focused on certain model organisms that are robust and easier to manipulate [14–16]. Common models for eukaryotes include sperm cells [17], *Chlamydomonas*, a single cell algae with two flagella, usually swimming in a breaststroke style [18–23], *Volvox*, a colony organism of thousands cells, with two cell types (somatic and germ cells), and *Xenopus* embryos as a model of the developmental biology in vertebrates [24].

A specific question of interest for many years has been the onset of collective dynamics in systems with many cilia: surface waves enable swimming or transport of fluid across surfaces [16]. Model systems for this are the somatic cells of *Volvox*, which have two flagella each. The strokes of cilia are coordinated across the colony, allowing it to swim and control direction [9, 25]. There are also microorganisms with many cilia per cell, for example *Paramecium*, another commonly studied example, is a single cell organism with a carpet of flagella on its exterior, again enabling it to swim [26–28]. Other well known organisms with many cilia include *Opalina*, *Stylonchia*, and other ciliated protists, while others exhibit cilia during a particular stage of the life cycle e.g. starfish larva [12, 29–31]. Some organisms, like the starfish larva, have a rudimentary central nervous system but others do not [32, 33]. Even ciliated organisms lacking a nervous system can exhibit multiple gaits, i.e. styles of swimming [34–37]. The strokes are usually related to feeding (chemotaxis), finding optimal light conditions (phototaxis), or evasion. It is thought that evolving some form of stress-response gait to evade danger is beneficial. Similarly some organisms respond to changes in nutrient or chemical levels, examples include *Chlamydomonas* which displays phototaxis [37] and some dinoflagellates which change swimming style depending on the density of prey [38, 39]. Metachronal (travelling waves) of small appendages are also seen in a variety of taxonomically diverse marine invertebrates—to our knowledge there has been no investigation of the role of hydrodynamic forces in coordinating metachronal waves in these species, but the surrounding hydrodynamic fields have recently been explored [5]. The diverse range of solutions that have evolved to combine feeding and locomotion control, through ciliary dynamics, raises the question if hydrodynamic forces play a role in development of gaits.

In a basic system like the active oscillators in simulations, all aspects of the fluid flow and of the oscillator coupling are under control. We implement three simple control mechanisms, loosely inspired by changes that could be actuated by ciliated organisms, into a system of 'rowers'. The changes we then see in the collective state of rowers do not prove that a more complex organism is changing gait in an equivalent way, that is not the purpose of the simple model. The purpose is to show, perhaps unexpectedly, that a very minimal and fundamental

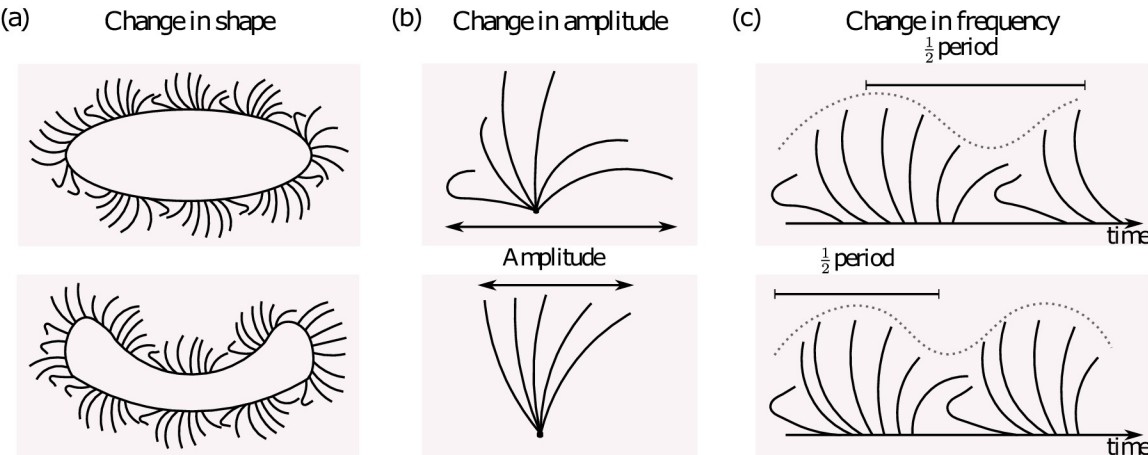

**Fig 1. There are simple mechanisms available to microorganisms to change coupling forces between cilia.** (a) The organisms could change shape, and so dramatically reduce the coupling between groups of cilia. (b) The amplitude of the cilium beat could be altered, which would in turn change the coupling strength between neighbours. (c) Similarly, the frequency of a cilium could be increased or decreased. Both the amplitude and frequency changes could either increase or decrease the coupling strength, depending on whether there is an increase or decrease in the parameter being changed.

modulation can lead to changes in collective dynamics. This will hopefully motivate future experiments to see whether or not gaits can be created and switched through hydrodynamic forces. The rower model is one of several types of active oscillator model that have been developed to study synchronisation in systems with fluid coupling in a low Reynolds' number regime [1]. These very simple models approximate the cilia with spheres, and can be realised experimentally with optical tweezers in small systems [40–42]. We use the rower model because of its simplicity in experimental and numerical implementations, and possibility of theoretical insight through mode analysis [1]. The alternative minimal models are 'rotors', which we investigated recently in [43]; both rowers and rotors produce general synchronisation features in viscously coupled systems. The two classes of models can in fact both be mapped onto the same coarse-grained description of pairwise phase interaction [44], so both can be tuned to exhibit the same collective properties. Previously, rower models have been used to investigate generic features of oscillator pairs or small systems [42, 44–48]. Questions concerning alignment, metachronal waves, and the effect of heterogeneity in spatial layout have all been studied in larger system using the model [45, 49–51]. Schematic illustrations of the mechanisms we explore are shown in Fig 1. We already know that these minimal models can be fine-tuned to replicate many aspects of cilia collective dynamics. The purpose of the study here with this basic model is to explore which changes in the minimal model, with purely viscous hydrodynamic interactions, might correspond to the complex swimming gait switching seen in biological systems. We show that subsets of model-cilia with specific phase locked dynamics can form, as groups of oscillators with phase-locking in opposing phase, and that these dynamical regimes can be switched (controlled) using the geometry of the system, or small alterations to some of the oscillators, as control parameters.

## Materials and methods

### The rower model

The rower model is a phase-free driven oscillator, it represents in a highly simplified fashion the dynamics of cilia [1]. Although biological flagella involve shape changes of slender

filaments [14, 52], each cilium is represented in the model by a single oscillator bead. In the far-field, i.e. for large separation distances, the flow field resulting from this sphere should match closely to that of a filament [1]. The complex internal dynamics of the active filament [53–56] are coarse-grained into the details of how the bead is moved through the fluid. The oscillations of the rower model are created by a geometrically updated driving potential. The potential must be either strictly increasing or decreasing, depending on whether the trap is attractive or repulsive. The switch is geometric i.e. configuration (position) based: When a bead passes a given threshold the driving potential is updated (flipped) and the rower reverses its direction [40]. To simulate the rowers, the position of each rower is updated using a Langevin equation, see Section 1 in S1 File for full details. The rowers are hydrodynamically coupled through the Blake tensor, which accounts for the presence of the wall. An illustration of the driving potential is shown in Fig 2b, highlighting that the resulting force is position dependent. At the scales considered, Brownian noise comes into play. We include the noise using the Ermack-McCammon method, further information is included in Section 1 in S1 File.

## Configuration to explore switching metachronal waves

**Geometric parameters of the rower chain.** Multi-ciliated organisms with many cilia are famous for their metachronal waves [29, 30, 34, 45, 57], they appear as a steady state in the dynamics, with a consistent phase difference between neighbouring cilia, i.e. a consistent time delay between each cilium's beat. Metachronal waves have been replicated before in systems of rowers [45, 57]. Taking inspiration from the work of [45], we use oscillators with a large

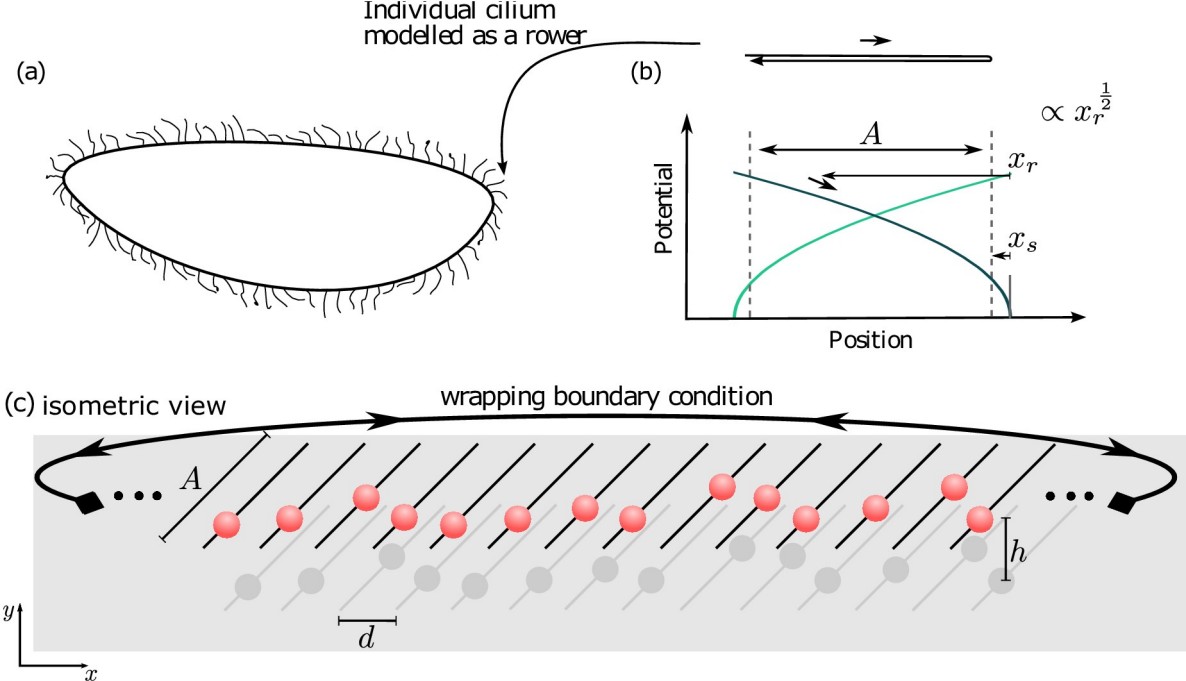

**Fig 2. The coordinated motion of cilia in multi-ciliated organisms can be modelled using arrays of driven rower oscillators.** (a) Schematic illustration of a multi-ciliated organism. (b) Each of the "hairs" on the surface, the cilia, is modelled as a driven bead. The beads are each driven by a potential trap with a position based switch, a set distance from the vertex $x_s$. The update of the active trap produces oscillations with amplitude $A$, and free phase an period. (c) The rowers oscillate in a plane at a height $h$ above a surface. The oscillations are 45˚ off the $x$ axis, with each trajectory separated by a distance $d$ in the $x$ direction. The whole system is wrapped in a periodic boundary condition to mimic the elliptical nature of an organism.

amplitude at high density, close to a surface, to represent microorganisms with carpets of cilia. Specifically we choose the amplitude $A/a = 30$, average rower separation $d/a = 7$, and height above the boundary $h/a = 3$. Each of these are measured in terms of the bead radius $a$. To allow this dense spacing with large oscillations, without risk of approaching near field (or overlaps!) the rowers are set to oscillate at 45° off the horizontal, with the distance between oscillators measured along the horizontal. The exact value of angle is an arbitrary choice; the exact values of these parameters are not meant to represent a particular biological system, that is not our purpose here. A schematic of this layout is shown in Fig 2c. The chain itself is 60 rowers long. Periodic boundary conditions are implemented to reproduce the features of the cilia carpet. Unlike in [45], we do not want to suppress changes in the direction of the metachronal wave. Consequently there is no additional symmetry breaking in the driving force for the system here.

**Simulation parameters for modelling inspired by starfish larvae.** The beads are driven by an attractive square root potential $k_x x_r^\alpha$, with $\alpha = 0.5$ and the switch point $x_s/a = 2.5$. This produces metachronal waves with long correlation length. The trap strength $k_x$ is selected to produce a period of 2s, with the step size $2 \cdot 10^{-3}$ cycles and the simulation length 2000 cycles. In each simulation the starting positions are drawn from a uniform distribution $\mathcal{U}(-A/2, A/2)$, and the trap randomly oriented. Each case is re-initialised 50 times, with new starting positions in each simulation. The dimensionless noise, as defined in Section 3 in S1 File, is set to $\xi = 3.7 \cdot 10^{-5}$.

## Posited control mechanisms

The organisms that inspire this work are relatively basic, with the simplest central nervous system only available to the most complex examples introduced above, e.g. the starfish larva. This restricts the control mechanisms that can be at play over the dynamics in the cilia carpet. We test the effect of control mechanisms based on simple changes, that could be exploited by even the simplest organisms: the idea of shape change, and small localised changes in the cilia beat.

**Shape change to modify hydrodynamic coupling directly.** It is feasible that small organisms can alter their shape in response to some external stimulus [36, 58]. This would directly change the hydrodynamic coupling between neighbouring groups of cilia. By bending and changing its shape, an organism could alter the orientation and distance between cilia clusters. The nature of the solid surfaces, and whether they are flat or curved, also enters the hydrodynamic coupling strength because of screening effects [45]. In the very minimal model studied here, the 1-dimensional chain can be thought of as a line of longitude from anterior to ventral, as the organisms typically are at least approximately of cylindrical symmetry. To mimic the shape change of organisms to an effect along the line of rowers, an additional distance separating some of the rowers is introduced. This is illustrated in Fig 3a. The extra distance $d_X$ is introduced regularly between every $N_g$ rowers. Only reduced coupling, i.e. $d_X > 0$, is considered in all cases.

**Cilia oscillation size.** Our second mechanism represents the possibility that the beat of the cilia themselves might be altered to change the swimming stroke of the organism. While there are still unanswered questions about the precise process underpinning the different aspects of cilia waveforms, it is accepted that the amplitude can be varied through some regulation of the dynein activity, perhaps most simply through control of ATP flow to the cilium [59]. Therefore a possible control mechanism stemming from the motion of the cilium is a change in the amplitude.

To investigate the consequences of this idea in a minimal fashion, we alter the amplitude of regularly spaced rowers. This is illustrated in Fig 3b. Every $N_g$ rowers, the amplitude is altered

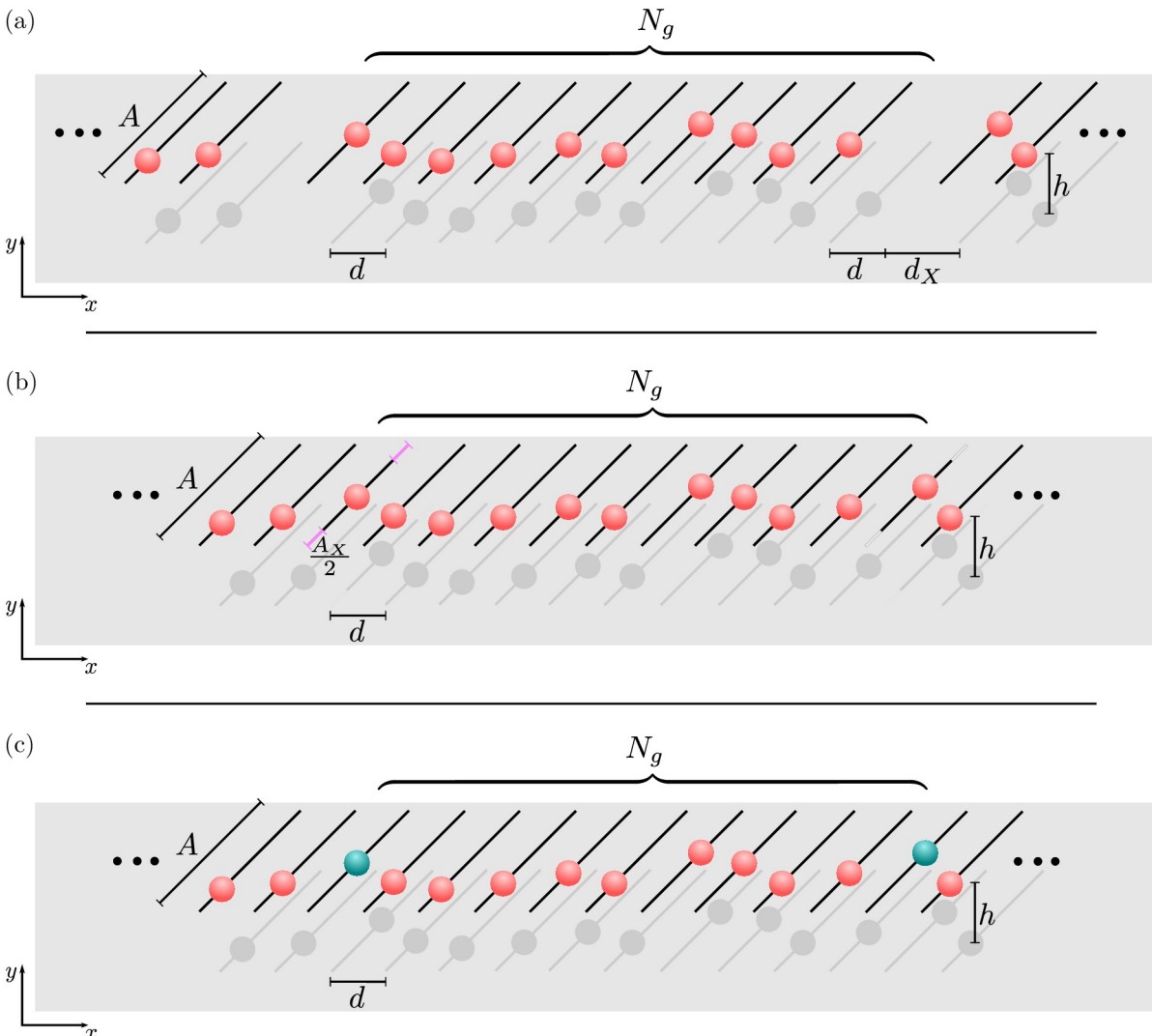

**Fig 3. Three possible control mechanisms for the formation of phase-locked subsets, the three control strategies explored to switch swimming gaits.** (a) Shape changes in a cell could alter to strength of coupling between groups of cilia, this is modelled by increasing the distance between groups of rowers the same size. Every $N_g$ rowers the spacing is increased by $d_X$, thereby changing the coupling between groups of size $N_g$. Similarly the beat of particular cilia could be altered to create changes in the gait. (b) The amplitude of the beat is altered every $N_g$ rowers by $A_X$. The amplitude can either be increased or decreased, but the frequency of the beat is maintained. (c) Alternatively, the amplitude of the beat can be kept constant while the frequency is varied, represented by the change in colour of the rower. In this way particular rowers can be moving faster or slower than the others, and drive changes in the chain.

by $A_X$, a factor which can be either positive or negative. The frequency of the rowers is maintained across the entire chain, so a rower with $A_X < 0$ has decreased trap strength. This in turns leads to a reduction in the driving force, and coupling strength, with the other rowers. Conversely, if $A_X > 0$ the bead velocity and the coupling strength increase. This is an additional freedom that is not considered when changing the distance between subsets.

**Cilia frequency.** The third aspect we consider is another possible effect stemming from regulating the dynein activity or altering the ATP concentrations: the beat frequency of the cilium might change. This is implemented in a similar way as the previous options, with the frequency of every $N_g$th rower scaled by a factor $f_X$. This is illustrated in Fig 3c, with the rowers with altered frequency marked in green. The frequency can be increased or decreased with

$f_X > 1$ or $f_X < 1$, which is implemented in the model by scaling the trap strength $k_x$ to alter the frequency. Consequently this either increases the coupling strength when $f_X > 1$, or decreases the coupling when $f_X < 1$.

## Results

### Behaviours resulting from the different mechanisms

The chain of rowers will settle into a steady phase-locked state if it is unperturbed and/or irregularities are small. When the regularity in the chain is disrupted, either through position, amplitude or frequency disorder, the phase profile of the rower chains can develop additional structure. Examples of the observed behaviour are shown in Fig 4.

Groups of $N_g$ rowers are formed when additional distance is included between the subsets. The groups can be recognised by the increase in the phase difference, and reversals that occur in the phase profile; a reversal is used to describe the situation where the phase difference changes sign, i.e. in one group the left neighbours lead, while in another the right neighbour would be ahead in the cycle. An example of the reversals is shown in Fig 4d, where $d_X/d = 16/7$ every 10 rowers. The phase profile is marked by the black line to emphasise its shape. A reversal appears as a 'chevron' in the phase profile. In this particular case there are three chevrons across the chain, each occurring at the point of disruption. The dotted black lines mark the rowers that have an increased separation distance. In terms of comparing with biological systems, see e.g. the patterns of beating cilia and flows in starfish larvae in [30], each linear part of a chevron maps to each region of the organism that shows coherence and a uniform phase-locked dynamics between neighbours.

Similar phase profiles appear when the amplitude is varied. When the amplitude change is small, the chevrons are sharp, like they are in the modulated-distance case. However, the placement of the chevrons is not as systematic: Not all reversals occur at the rower with altered amplitude. This is shown in Fig 4e. One point of the chevron is at the 20th rower, which is altered. A second reversal in the profile is necessary to maintain the periodic boundary conditions. This point occurs in what would be expected to be the centre of a subset, around the 55th rower in the current labelling. Changing the amplitude further leads to additional differences in the shape of the profile. Where previously the reversals in the phase profile were sharp, and the phase difference consistent in size if not direction, now the profile is smoother with the phase difference varying between and in subsets. This is shown in Fig 4f, which shows a more sinusoidal pattern than the sharp zigzag example of Fig 4d.

Varying the frequency of the rowers creates similar subsets, but with some features unique to the style of disruption. Increasing the frequency of every $N_g$th rower appropriately appears to stabilise the in-phase subsets. These subsets are seen regularly in the frequency-modulation case, with each in-phase subset separated by a phase-locked subset. An example of such behaviour is plotted in Fig 4h. For larger changes, the altered rowers do not phase-lock with the others. When the change is a frequency reduction, the other rowers phase-lock, with the altered ones remaining incoherent. This often manifests in the profile as 6 small kinks, see Fig 4g. If the altered rowers have a frequency that is too high, the system is unable to stabilise. The asymmetrical effect will most likely stem from the difference in coupling strength, as low frequency rowers have lower coupling strength allowing the remaining rowers to coordinate.

### Consistency of the observed phase profiles

**Modelling chevron occurrence using a binomial distribution.** The shape profile exhibited is dependent on the initial configuration of the system. It is also important to consider the size of the basin of stability for a given behaviour, not just the features that can occur. To that

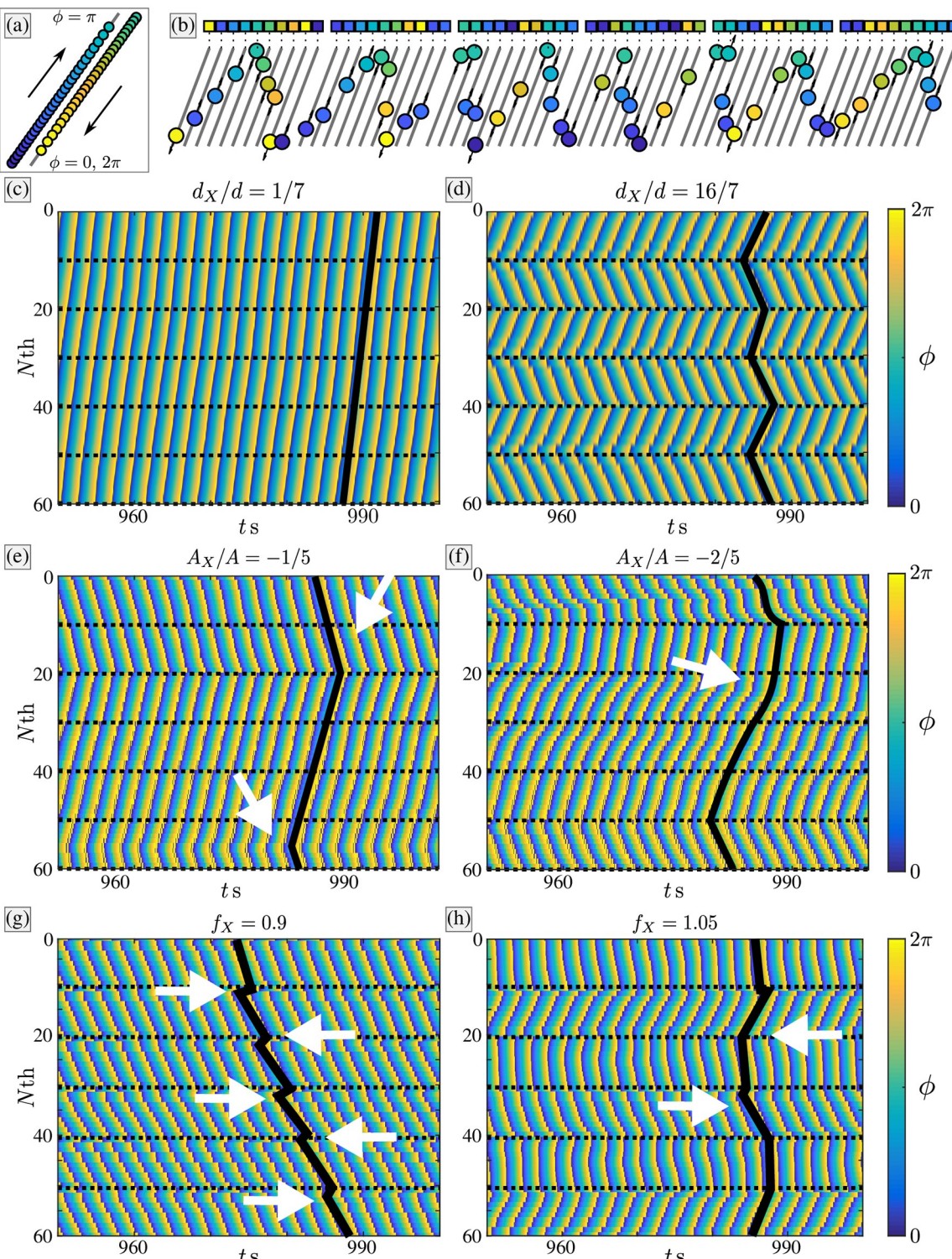

**Fig 4. Examples of subset formation with the different control methods: Behaviours in the chain are measured using a proxy variable for their position and direction.** (a) The position and direction of a rower can be measured by a single variable, the phase, represented by the blue-yellow gradient. (b) A chain of rowers, colour coded by their phase. The single variable means each rower can be represented by a single value, and compressed onto a single line. This is represented by the coloured square above each trajectory. The one-dimensional nature of the representation allows the system to be tracked over time. (c) The phase, represented by colour of a chain of rowers over time. In this case all the rowers are near one another, and the chain settles into one global phase-locked state. The black line highlights its structure. (d) Increasing the separation creates subsets, which can have opposing directions for the phase difference. Similar features are observed in the amplitude modulation case, but the subsets are less consistent. This can manifest as in (e), where the

phase reversal does not occur at the rower with altered amplitude. (f) For larger changes, the phase profile becomes smoother. The phase profile generally has a different structure when the frequency is changed. For large differences in frequency the altered rowers do not phase-lock, while the rest coordinate around them. This is shown in (g), where each altered rower produces a kink in the profile. When the rowers are only slightly accelerated, they are likely to form in-phase subsets, with alternating subsets phase-locked between them, as in (h).

end, the consistency of observing the reversals in the phase profile is modelled using a binomial distribution. The process to measure the chevron formation involves measuring the prominence of terms in the Fourier series of the phase profile. This provides a snapshot of the dominant features in the phase profile, at a given time. We are only interested in large features that occur over several rowers, so we discard any Fourier modes that are smaller than three rowers in scale; $N/(2k) > 3$, where $k$ is the number of the Fourier mode, which for $N = 60$ gives $k < 10$. The largest Fourier coefficient should correspond to the overall structure of the phase profile. An example of the process is shown in S1 Fig in S1 File. In the event that all the coefficients are smaller than 0.15, the structure in the phase profile is considered too small to indicate a chevron. This threshold of 0.15 was set following inspection of several chevron and no chevron cases. If a reversal has occurred the largest coefficient is a crude approximation of the phase difference between pairs. Consequently the threshold is akin to the phase difference expected when every 40th rower is disrupted ($2\pi/40 \approx 0.15$) and is unlikely to be a feature in chains of 60 rowers.

The assumptions underpinning the binomial distribution are to be kept in mind when considering the number of reversals in the phase profile. First, the requirement that there are only two outcomes is easily met by either the existence or non-existence of a reversal. Second, the independence of each trial is likely to be met if the subsets are large enough. However, if chevron formation is not independent, that would be notable on its own. It is the final assumption, the number of trials being fixed, that may be a barrier to using the binomial model. This is unlikely to be a problem in the amplitude and distance modulation cases, which generally showed reversals at the point of disruption. The frequency disruption, however, has more non-uniformity in its behaviour and is unlikely to meet the assumption. This will become apparent later when discussing the frequency results.

The probability of chevron occurring, $p_{ch}$, is measured by dividing the total number of chevrons observed by the total possible number chevron occurrences; reversals must occur in pairs to satisfy the boundary conditions, so with six points of disruptions a maximum of 3 chevrons is expected. With 50 simulations of each level of disruption, the maximum number of chevrons expected to occur is 150. This measure assumes that each chevron forms independently in the chain. Example distributions of the fitted binomial, and the data it is based on, are also included in S1 Fig in S1 File.

**Extra distance.**    The subsets formed exclusively in groups of $N_g$ rowers, separated by the additional distance $d_X$. This implies the number of chevrons that can occur in any one chain is three. The periodic boundary conditions mandate that the phase difference wraps, consequently each $\wedge$ shape in the chevron has a corresponding $\vee$. The maximum number of chevrons fixes the number of trials in the binomial modelling. Initially there are no subsets, and we expect the probability of a chevron occurring to be 0. At the opposite extreme, once the groups are far enough apart they should decouple completely, so the probability of a reversal should be random, i.e. 1/2. It is in the intermediate regime that a maximum in chevron probability is observed. This indicates that there is a choice of distances in which the chance of creating subsets with opposing directions is preferred.

The observed probability for the number of chevrons in a chain is shown in Fig 5. The distribution of the results are plotted in Fig 5a, with the random case ($p_{ch} = 0.5$) overlaid as

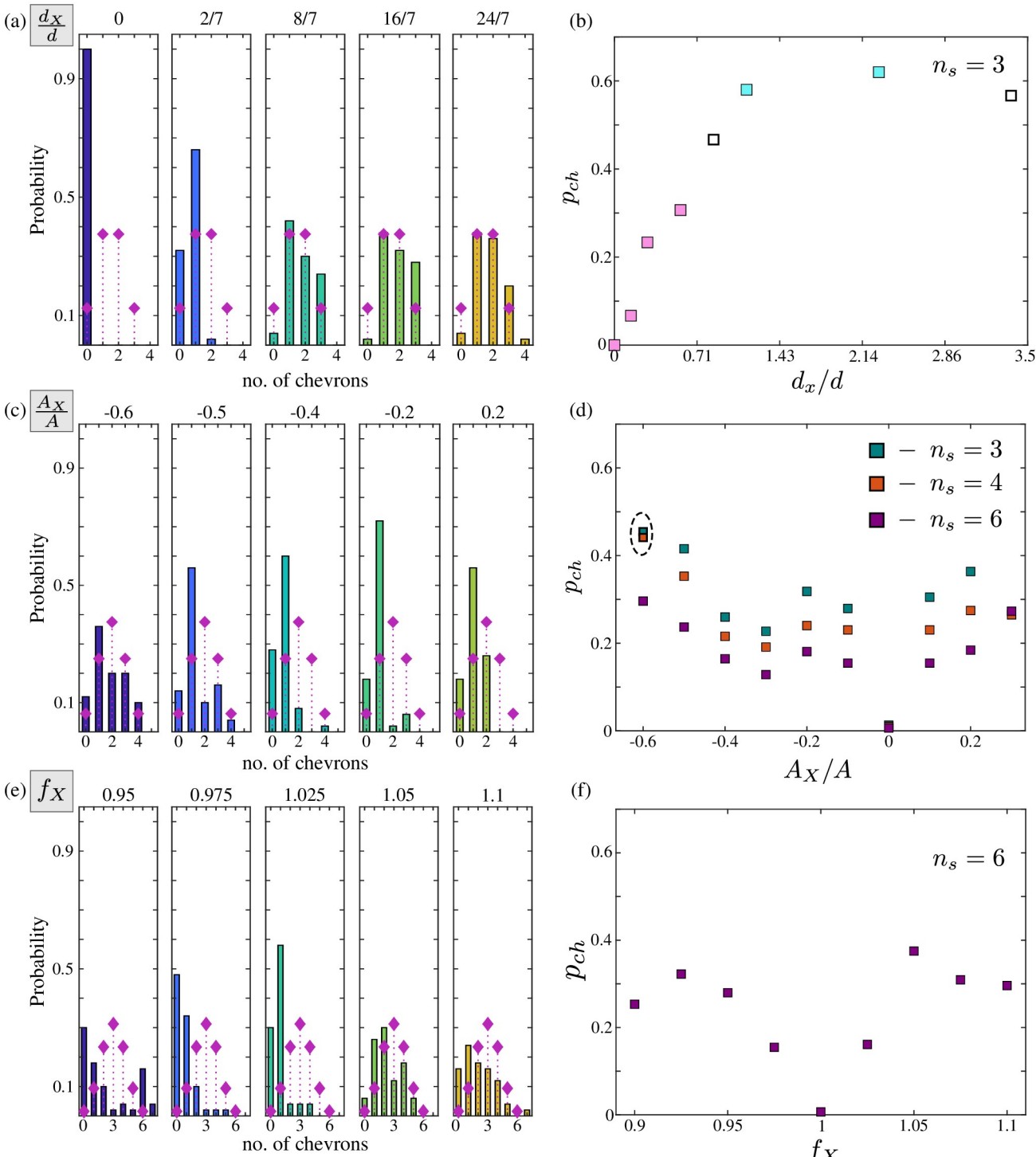

**Fig 5. All control strategies can create reversals in the chain, but with varying levels of success.** Decreasing the coupling between sections of the chains increases the probability of a reversal in direction. Extra spacing causes the most consistent effect, but increasing the frequency is unique in causing groups of in-phase rowers. (a) The number of chevrons observed in each chain for different spacing, with the no-preference case of $p_{ch} = 0.5$ overlaid (♦) for reference. When there is an additional $8\mu m-16\mu m$ of separation, there are more than three chevron cases in excess of what would be expected by chance. (b) The probability of observing a chevron as the separation is changed. The coloured markers are the cases where some preference for chevrons was demonstrated (two-tailed; $p < 0.05$). The pink markers indicate a reduced chance of a chevron occurring, blue indicates the number of chevrons exceeds what would be expected by chance. (c) The same probability distributions, but for increasing $A_X$. There are now cases with more than three chevrons, revealing a decrease in control in the reversal positions. Here the overlaid distributions are $Bi\hat{A}$ (4, 0.5). (d) The probability of success as $A_X$ is varied. There

are three choices for the maximum number of chevrons possible in any given chain, with $n_s$ = 3, 4, 6 indicated by green, orange and purple markers respectively. Cases 3 and 6 are for comparison with the other results, while $n_s$ = 4 reflects the maximum number observed. The two circled markers are the two cases where the results could reasonably be drawn from $Bi \sim (n_s, 0.5)$. (e) The chevron distributions when varying frequency. The ($\blacklozenge$) now represent the distribution $Bi \sim (6, 0.5)$. Rowers with large detuning values do not synchronise or stabilise, undermining the binomial assumption. These either lead to bimodal or uniform distribution depending on the behaviour of the remaining rowers. (f) Despite these problems, the binomial probability is calculated and plotted against the detuning $f_X$. The binomial probability may be valid for small values of detuning, but a high probability of chevron formation is never observed.

diamond markers. When $d_X$ = 0 chevrons are never observed, but as $d_X$ is increased they become more common. Of particular interest are the cases where $d_X$ = 8/7 and 16/7, where the number of cases with chevrons exceeds what would be expected by chance. This is corroborated in Fig 5b, where the probability of observing a chevron, $p_{ch}$, is plotted against $d_X$. The markers left uncoloured do not have enough evidence to assume some preference for either chevrons or no-chevrons (two-tailed; $p > 0.05$). The pink and blue markers are for cases that do exhibit some preference, with the two categorised by inspection. Pink markers indicate states that have a decreased probability of chevrons, and blue a better than random chance.

**Changing oscillation amplitude.** The changes in amplitude lead to some chevrons occurring in the centre of an expected subset. This undermines the assumption that there are three trials, because the reversals are no longer restricted to the $N_g$th rowers. Instead, three options are considered, with the maximum number of chevrons in any one chain $n_s$ assumed to be either 3, 4, or 6. The case where $n_s$ = 3 is to mimic the previous case of distance modulation; $n_s$ = 4 is the maximum number of clear chevrons observed; and $n_s$ = 6 assumes a chevron can appear at the amplitude-altered rowers, or half way between them. The distributions for the number of observed chevrons is shown in Fig 5c. Here the overlaid diamond markers are the expected number of observations by random chance when there are 4 possible locations for a chevron. Across the different cases, there is never an abundance of chevrons appearing, with one chevron usually the most likely outcome. When the amplitude is greatly reduced, i.e. $A_X/A$ = −0.6, the distribution most closely resembles what would be expected for $p_{ch}$ = 0.5, but still demonstrates a skew towards one reversal. The probability of a chevron assuming the different number of chevron locations is shown in Fig 5d. The green markers are for 3 locations, the orange for 4, and purple when $n_s$ = 6. Unsurprisingly, increasing the number of trials reduces the observed probability. Regardless of the trial number, none of the changes in amplitude demonstrates a preference beyond that of chance for chevron formation. Further, the only case that has apparent random chevron occurrence is $A_X/A$ = −0.6.

**Frequency alterations to regularly spaced rowers.** The structure in the phase profile is often less regular when the frequency of the $N_g$th rower is strongly altered. When the frequency is lowered, the detuned rowers fail to phase-lock but the remaining rowers synchronise. This appears as a sixth-order term in the Fourier series, and is a lack of coordination rather than the appearance of six chevrons. An example can be seen in Fig 5e. The case where $f_X$ = 0.95 is clearly bimodal centred on 0 or 6 chevrons, and is demonstrably not well represented by a bimodal distribution. At the other extreme, when the adjusted frequency it too large, the chain does not stabilise, and the most prominent Fourier term also does not correspond to a stable chevron state. This manifests as a flattening of the distribution, as seen when $f_X$ = 1.1 in Fig 5e. Nonetheless, the probability of observing a chevron is calculated assuming that a total of six can appear, see Fig 5f. Despite the irregularity in doing so, a strong positive skew in the distribution would still be expected to produce an uptick in the probability. This does not appear when plotting the probability against the detuning level.

## Conclusion

Three mechanisms were chosen as simple and distinct ways to alter the hydrodynamic coupling in arrays of rowers that mimic systems with many cilia. This is a very idealized system that intends to explore a possible connection to gait switching in viscous swimming and fluid propulsion. These modifications to coupling could be plausible control mechanisms in ciliated mircoorganisms, but there is no experimental evidence so far. In the model, the inclusion of additional separations between some cilia leads to the greatest control over the emergent dynamics state, with exact grouping of the coordinated rowers. Most likely this is the result not only of the neighbouring rowers having decreased coupling, but of all the rowers in a given subset having a weaker coupling versus the other subgroups. In contrast, altering aspects of the oscillation leads to subgroups with a wider range of phase differences, and, in the case of frequency modulation, has demonstrated that in-phase subsets can be stabilised. However, for both the amplitude and frequency alterations, the resulting subsets were less consistent and often occurred over a small parameter window. All three mechanisms demonstrate that the formation of coordinated subsets can be engineered, so that previously unstable states can become preferable under appropriate modulated conditions.

The phase reversals in the model system we study would correspond to a change in swimming velocity, or direction, or a change in the patterns of flow to affect foraging in a ciliated organism. The results here suggest that a swimming micro-organism could for example modulate its shape to assist in grouping subsets of cilia into particular phase-locking, and thus switch swimming gait. Shape-change would decrease the coupling between cilia, in a similar way as increasing the spacing between rowers. The modulation of separation spacing provided the most control over the location of the chevrons, which would be important for vortex formation in the fluid velocity. Additionally, decreasing the coupling in this way led to the sharpest chevron formation. The effect may also be relevant when considering cilia on the surface of coral, where the bumpy surface is a potential explanation for the flow reversals that create vortices, vital for mixing and waste expulsion within the coral [60]. This is not the first suggestion that coral structures can enhance the mixing within coral [61]. Along a similar vein, the results on frequency modulation suggest that if some cilia accelerate their beating, then a patch would coordinate and beat in-phase. This could be relevant for *Paramecium*, which responds to moderate threats with a coordinated in-phase beat, postulated to depend on calcium signalling [34, 62]. These are just two examples we are aware of; with such a breadth of multiciliated organisms in nature [12] we expect many further investigations into switching swimming gaits in microorganisms to unearth the balance of the physical perturbations modelled here, versus the biochemical intracellular regulation.

## Supporting information

**S1 File.**
(PDF)

## Acknowledgments

We would like to thank Raymond Goldstein for useful discussions.

## Author Contributions

**Conceptualization:** Evelyn Hamilton, Pietro Cicuta.

**Formal analysis:** Evelyn Hamilton, Pietro Cicuta.

**Funding acquisition:** Pietro Cicuta.

**Investigation:** Evelyn Hamilton.

**Methodology:** Evelyn Hamilton.

**Resources:** Evelyn Hamilton.

**Software:** Evelyn Hamilton.

**Supervision:** Pietro Cicuta.

**Visualization:** Evelyn Hamilton.

**Writing – original draft:** Evelyn Hamilton, Pietro Cicuta.

**Writing – review & editing:** Evelyn Hamilton, Pietro Cicuta.

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
