## [Decision Letter · Decision Letter 0]

18 Jan 2021

PONE-D-20-33443

A simple model for switching of swimming gaits in microswimmers, by geometric control of cilia synchronization

PLOS ONE

Dear Prof.Cicuta,

Thank you for submitting your manuscript to PLOS ONE. After careful consideration, we feel that it has merit but does not fully meet PLOS ONE’s publication criteria as it currently stands. Therefore, we invite you to submit a revised version of the manuscript that addresses the points raised during the review process.

One of the original referees is now happy with the manuscript. However, an additional referee has raised a series of important points that need to be addressed before publication. 

We look forward to receiving your revised manuscript.

Kind regards,

Tom Waigh

Academic Editor

PLOS ONE

Journal Requirements:

Reviewers' comments:

Reviewer's Responses to Questions

**Comments to the Author**

1. Is the manuscript technically sound, and do the data support the conclusions?

Reviewer #1: Yes

Reviewer #2: Partly

2. Has the statistical analysis been performed appropriately and rigorously? 

Reviewer #1: N/A

Reviewer #2: Yes

3. Have the authors made all data underlying the findings in their manuscript fully available?

Reviewer #1: Yes

Reviewer #2: Yes

4. Is the manuscript presented in an intelligible fashion and written in standard English?

Reviewer #1: Yes

Reviewer #2: Yes

5. Review Comments to the Author

Reviewer #1: The authors responded to all comments. I can now recommend the manuscript for publication.

Reviewer #2: This article proposed a theoretical model for metachronal waves on the surface of ciliated organisms. The model is an extension of a versatile and insightful rower model employed by Cicuta and collaborators, in a number of recent publications. The suggestion that organisms can possibly control the phase delay between nearby cilia and the nature of metachronal waves by changing certain physical parameters associated with ciliary arrays is indeed an intriguing and sensible idea.

However, the present model itself is highly idealised, both in terms of the dynamics of the individual rowers, and in their arrangement in a linear one-dimensional chain with wrapped boundary conditions. I believe additional work will be required to extend this to make it biologically relevant. The authors did not attempt to connect the synchronization patterns they obtained to swimming dynamics, and provided no evidence (either theoretical or experimental) to suggest that transitions in the patterns of such oscillators can actually reverse or significantly alter the fluid landscape around the ciliated swimmer. The association with ‘swimming gaits’ in real organisms is at best tenuous, and at worst misleading.

In my view the authors have two options towards publication, either revise the paper to make it clear this is a theoretical model looking at synchronization phenomena in a 1D chain of oscillators designed to mimic cilia (i.e. do not refer to ‘swimming gaits’ in the title, tone down the claims of abstract – which are not supported by the results of their current model, and reserve all discussion of how they think this could translate in real organisms to the discussions section, which they could expand to include free-swimming), or else provide further experimental or theoretical evidence that ‘swimming’ will be affected by the types of perturbations considered in this work.

key comments.

1. Stronger justification for why the rower models ‘are expected to produce general synchronisation features in viscously coupled systems’. This is a very general statement. The authors should explain how the rower model performs compared to other commonly used cilia models, and what are its advantages and limitations.

2. The authors consider the dynamics of a chain of rowers – but this is far from a 2D carpet or a ciliated surface. Why should this very idealised geometry be expected to represent what happens in the cell? Why the choice of 45 degree incline – seems arbitray, what is the motivation for this set-up?

3. Another point about choice of parameters. This is not well explained – for different organisms exhibit very different metachronal wave properties, differ in cilia length, spacing between cilia etc. The authors appear to be inconsistent throughout about which class of organisms their work is modeling – they mention simulation parameters inspired by starfish larvae, yet the drawings and discussions focus on paramecium (much smaller scale)? Then in the conclusions they discuss corals? Their work is clearly more relevant for singular rings of cilia (found in some organisms), rather than sheets.

4. According to the operational principle of most cilia, beat frequency and amplitude will be strongly coupled – it’s unlikely that one will be varying but not the other. Therefore distinguishing between control mechanisms b & c seems to be somewhat problematic? Can this be better justified? (I refer back to 1.)

5. Are the authors aware of any examples of metachronal wave reversal in real organisms induced by a change in separation between some of the cilia? They mention the starfish larvae example – but it is unclear if those patterns have anything to do with shape changes? Or frequency? What about this idea that some level of disorder in the spacing could help with control the generation of locally synchronized subsets (lines 306/307) – is there any evidence from the literature? (Note that the well-documented gaits in paramecium are unlikely to result from a mechanism such as described by the present model. In paramecium, and no doubt other species, the switch occurs due to some rapid cellular signalling, which abruptly halts or reverses the direction of ciliary beating. )

6. The analysis of the chevron profiles using statistical methods – though interesting, it is unclear how this would relate to swimming. Are the authors suggesting that phase reversals are associated with reversals in the actual swimming direction?

7. One suggestion would be to compare flow pumping by these chains of rowers – this would still be far from a force/torque free swimmer, but would be more relevant to biological systems. See for example https://www.pnas.org/content/117/48/30201.short.

8. There are many ways to control swimming gait using cilia, hydrodynamics is just one possibility. In fact, many more authors have studied non-hydrodynamic mechanisms for changing the swimming trajectory, these should be discussed. See for example studies on paramecium escape reactions, or even the steering gaits of uniflagellate sperm which are example of shape changes.

Other:

Line 41: Ciliophora is not a genus, and the term is being phased out

Line 43: Central nervous system – missing word

Fig 1 - what is the purpose of the green/pink color scheme? This could be confusing as panels of the same color don’t match up.

Fig 2- panel c - the drawing does not really make it clear all the rowers are on the same plane, and that this plane is above a wall. Make the plane more 3d?

Fig 4 – what is Ng here, presumably 10? This should be stated explicitly.

6. PLOS authors have the option to publish the peer review history of their article (what does this mean?). If published, this will include your full peer review and any attached files.

Reviewer #1: No

Reviewer #2: No

---

## [Decision Letter · Decision Letter 1]

11 Mar 2021

Changes in geometrical aspects of a simple model of cilia synchronization control the dynamical state, a possible mechanism for switching of swimming gaits in microswimmers

PONE-D-20-33443R1

Dear Dr. Cicuta,

We’re pleased to inform you that your manuscript has been judged scientifically suitable for publication and will be formally accepted for publication once it meets all outstanding technical requirements.

Kind regards,

Tom Waigh

Academic Editor

PLOS ONE

Additional Editor Comments (optional):

Reviewers' comments:

Reviewer's Responses to Questions

**Comments to the Author**

1. If the authors have adequately addressed your comments raised in a previous round of review and you feel that this manuscript is now acceptable for publication, you may indicate that here to bypass the “Comments to the Author” section, enter your conflict of interest statement in the “Confidential to Editor” section, and submit your "Accept" recommendation.

Reviewer #2: All comments have been addressed

2. Is the manuscript technically sound, and do the data support the conclusions?

Reviewer #2: Yes

3. Has the statistical analysis been performed appropriately and rigorously? 

Reviewer #2: Yes

4. Have the authors made all data underlying the findings in their manuscript fully available?

Reviewer #2: Yes

5. Is the manuscript presented in an intelligible fashion and written in standard English?

Reviewer #2: Yes

6. Review Comments to the Author

Reviewer #2: (No Response)

7. PLOS authors have the option to publish the peer review history of their article (what does this mean?). If published, this will include your full peer review and any attached files.

Reviewer #2: No

---

## [Editor Report · Acceptance letter]

23 Mar 2021

PONE-D-20-33443R1 

Changes in geometrical aspects of a simple model of cilia synchronization control the dynamical state, a possible mechanism for switching of swimming gaits in microswimmers 

Dear Dr. Cicuta:

I'm pleased to inform you that your manuscript has been deemed suitable for publication in PLOS ONE. Congratulations! Your manuscript is now with our production department. 

Kind regards, 

on behalf of

Dr Tom Waigh 

Academic Editor

PLOS ONE